# Hierarchical Neural Architecture Search for Deep Stereo Matching

*Xuelian Cheng[1,5], *Yiran Zhong[2,6], Mehrtash Harandi[1,7], Yuchao Dai[3], Xiaojun Chang[1],
Tom Drummond[1], Hongdong Li[2,6], Zongyuan Ge[1,4,5]
[1]Faculty of Engineering, Monash University, [2]Australian National University,
[3]Northwestern Polytechnical University, [4]eResearch Centre, Monash University,
[5]Airdoc Research Australia, [6]ACRV, [7]Data61, CSIRO
{xuelian.cheng, mehrtash.harandi}@monash.edu,
{xiaojun.chang, tom.drummond, zongyuan.ge}@monash.edu,
{yiran.zhong, hongdong.li}@anu.edu.au, daiyuchao@nwpu.edu.cn

## Abstract

To reduce the human efforts in neural network design, Neural Architecture Search (NAS) has been applied with remarkable success to various high-level vision tasks such as classification and semantic segmentation. The underlying idea for the NAS algorithm is straightforward, namely, to enable the network the ability to choose among a set of operations (*e.g.,*convolution with different filter sizes), one is able to find an optimal architecture that is better adapted to the problem at hand. However, so far the success of NAS has not been enjoyed by low-level geometric vision tasks such as stereo matching. This is partly due to the fact that state-of-the-art deep stereo matching networks, designed by humans, are already sheer in size. Directly applying the NAS to such massive structures is computationally prohibitive based on the currently available mainstream computing resources. In this paper, we propose the first *end-to-end* hierarchical NAS framework for deep stereo matching by incorporating task-specific human knowledge into the neural architecture search framework. Specifically, following the gold standard pipeline for deep stereo matching (*ie.,*, feature extraction – feature volume construction and dense matching), we optimize the architectures of the entire pipeline jointly. Extensive experiments show that our searched network outperforms all state-of-the-art deep stereo matching architectures and is ranked at the top 1 accuracy on KITTI stereo 2012, 2015 and Middlebury benchmarks, as well as the top 1 on SceneFlow dataset with a substantial improvement on the size of the network and the speed of inference. The code is available at LEAStereo.

## 1 Introduction

Stereo matching attempts to find dense correspondences between a pair of rectified stereo images and estimate a dense disparity map. Being a classic vision problem, stereo matching has been extensively studied for almost half a century [1]. Since MC-CNN [2], a large number of deep neural network architectures [3, 4, 5, 6] have been proposed for solving the stereo matching problem. Based on the adopted network structures, existing deep stereo networks can be roughly classified into two categories: **I.** *direct regression* and **II.** *volumetric* methods.

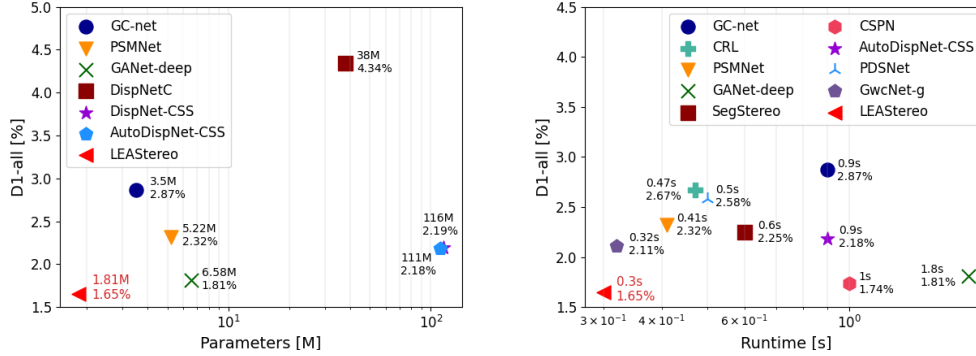

Figure 1: Our proposed method, LEAStereo (Learning Effective Architecture Stereo), sets a new state-of-the-art on the KITTI 2015 *test* dataset with much fewer parameters and much lower inference time.

Direct regression methods are based on direct regression of dense per-pixel disparity from the input images, without taking into account the geometric constraints in stereo matching [7]. In the majority of cases, this is achieved by employing large U-shape encoder-decoder networks with 2D convolutions to infer the disparity map. While enjoying a fully data-driven approach, recent studies raise some concerns about the generalization ability of the direct regression methods. For example, the DispNet [3] fails the random dot stereo tests [8].

In contrast, the volumetric methods leverage the concept of semi-global matching [9] and build a 4D feature volume by concatenating features from each disparity-shift. To this end, the volumetric methods often make use of two building blocks, the so-called **I.** *feature net* and **II.** *matching net*. As the names imply, the feature net extracts features from the input images and the matching net compute matching costs from the 4D feature volume with 3D convolutions. Different designs of the feature net and the matching net form variants of the volumetric networks [4, 10, 11, 12, 6]. Nowadays, the volumetric methods represent the state-of-the-art in deep stereo matching and top the leader-board across different benchmark datasets. Despite the success, designing a good architecture for volumetric methods remains an open question in deep stereo matching.

On a separate line of research and to reduce the human efforts in designing neural networks, Neural Architecture Search (NAS) has mounted tremendous successes in various high-level vision tasks such as classification [13, 14, 15], object detection [16, 17], and semantic segmentation [18, 19, 20]. Connecting the dots, one may assume that marrying the two parties, *ie.,*employing NAS to design a volumetric method for stereo matching, is an easy ride. Unfortunately this is not the case. In general, NAS needs to search through a humongous set of possible architectures to pick the network components (*e.g.,*the filter size of convolution in a certain layer). This demands heavy computational load (early versions of NAS algorithms [21, 22] need thousands of GPU hours to find an architecture on the CIFAR dataset [23]). Add to this, the nature of volumetric methods are very memory hungry. For example, the volumetric networks in [4, 11, 8, 24] require six to eight Gigabytes of GPU memory for training per batch! Therefore, end-to-end search of architectures for volumetric networks has been considered prohibitive due to the explosion of computational resource demands. This is probably why, in the only previous attempt[1], Saikia *et al.* [25] search the architecture based on the direct regression methods, and they only search partially three different cell structures rather than the full architecture.

In this paper, we leverage the volumetric stereo matching pipeline and allow the network to automatically select the optimal structures for both the Feature Net and the Matching Net. Different from previous NAS algorithms that only have a single encoder / encoder-decoder architecture [18, 25, 20], our algorithm enables us to search over the structure of both networks, the size of the feature maps, the size of the feature volume and the size of the output disparity. Unlike AutoDispNet [25] that only searches the cell level structures, we allow the network to search for both the cell level structures and the network level structures, *e.g.,*the arrangement of the cells. To sum up, we achieve the first *end-to-end* hierarchical NAS framework for deep stereo matching by incorporating the geometric knowledge into neural architecture search. We not only avoid the explosive demands of computational resources in searching architectures, but also achieve better performances compared to naively searching an architecture in a very large search space. Our method outperforms a large set of state-of-the-art algorithms on various benchmarks (*e.g.,*topping all previous studies on the KITTI and Middlebury

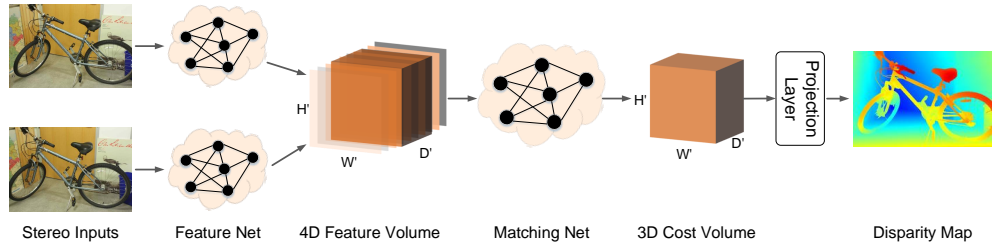

Stereo Inputs     Feature Net    4D Feature Volume    Matching Net    3D Cost Volume    Disparity Map

Figure 2: **The pipeline of our proposed stereo matching network.** Our network consists of four components: 2D Feature Net, 4D Feature Volume, 3D Matching Net, and Projection Layer. Given a pair of stereo images, the Feature Net produces feature maps that are processed by the Matching Net to generate a 3D cost volume. The disparity map can be projected from the cost volume with soft-argmin operation. Since the Feature Net and the Matching Net are the only two modules that contain trainable parameters, we utilize the NAS technique to select the optimal structures for them.

benchmarks[2] ). This includes man-designed networks such as [24, 26] and the NAS work of Saikia *et al.* [25], not only in accuracy but also in inference time and the size of the resulting network (as shown in Figure 1).

## 2   Our Method

In this section, we present our end-to-end hierarchical NAS stereo matching network. In particular, we have benefited from decades of human knowledge in stereo matching and previous successful handcrafted designs in the form of priors towards architecture search and design. By leveraging task-specific human knowledge in the search space design, we not only avoid the explosion demands of computational resources in searching architectures for high resolution dense prediction tasks, but also achieve better accuracy compared to naively searching an architecture in a very large search space.

### 2.1   Task-specific Architecture Search Space

We recall that the deep solutions for dense prediction (*e.g.,*semantic segmentation, stereo matching), usually opt for an encoder-decoder structure [18, 25, 20]. Inspired by the Auto-DeepLab [18] for semantic segmentation, we propose a two-level hierarchical search that allows us to identify both cell-level and network-level structures[3]. Directly extending ideas from semantic segmentation might not necessarily lead to viable solutions for stereo matching. A fully data-driven U-shape encoder-decoder network is often hard to train, even with the help of NAS [25] in regressing disparity maps. Volumetric stereo matching methods offer faster convergence and better performance as their pipeline makes use of inductive bias (*ie.,*human knowledge in network design). To be specific, volumetric solutions first obtain a matching cost for all possible disparity levels at every pixel (based on the concepts of 3D geometry) and then use it to generate the disparity map (*e.g.,*by using a soft-argmin operation). One obvious drawback here is the overwhelming size of the resulting network. This, makes it extremely difficult, if not impossible, to use volumetric solutions along the NAS framework.

In this work, we embed the geometric knowledge for stereo matching into our network architecture search. Our network consists of four major parts: a 2D feature net that extracts local image features, a 4D feature volume, a 3D matching net to compute and aggregate matching costs from concatenated features, and a soft-argmin layer that projects the computed cost volumes to disparity maps. Since only the feature net and the matching net involve trainable parameters, we leverage NAS technique to search these two sub-networks. The overall structure of our network is illustrated in Figure 2. More details about the cell and network level search are presented below.

#### 2.1.1   Cell Level Search Space

A cell is defined as a core searchable unit in NAS. Following [27], we define a cell as a fully-connected directed acyclic graph (DAG) with $\mathcal{N}$ nodes. Our cell contains two input nodes, one output node

and three intermediate nodes. For a layer $l$, the output node is $\mathcal{C}_l$ and the input nodes are the output node of its two preceding layers (*ie.*,$\mathcal{C}_{l-2}, \mathcal{C}_{l-1}$). Let $\mathcal{O}$ be a set of candidate operations (*e.g.,*2D convolution, skip connection). During the architecture search, the functionality of an intermediate node $s^{(j)}$ is described by:

$$s^{(j)} = \sum_{i \rightsquigarrow j} o^{(i,j)}\left(s^{(i)}\right) . \tag{1}$$

Here, $\rightsquigarrow$ denotes that node $i$ is connected to $j$ and

$$o^{(i,j)}(\boldsymbol{x}) = \sum_{r=1}^{\nu} \frac{\exp\left(\alpha_r^{(i,j)}\right)}{\sum_{s=1}^{\nu} \exp\left(\alpha_s^{(i,j)}\right)} o_r^{(i,j)}(\boldsymbol{x}) , \tag{2}$$

with $o_r^{(i,j)}$ being the $r$-th operation defined between the two nodes. In effect, to identify the operation connecting node $i$ to node $j$, we make use of a mixing weight vector $\boldsymbol{\alpha}^{(i,j)} = (\alpha_1^{(i,j)}, \alpha_2^{(i,j)}, \cdots, \alpha_\nu^{(i,j)})$ along a softmax function. At the end of the search phase, a discrete architecture is picked by choosing the most likely operation between the nodes. That is, $o^{(i,j)} = o_{r^*}^{(i,j)}$; $r^* = \arg\max_r \alpha_r^{(i,j)}$. Unlike [27, 25], we only need to search one type of cells for the feature and matching networks since the change of spatial resolution is handled by our network level search. DARTS [27] has a somehow inflexible search mechanism, in the sense that nodes $\mathcal{C}_{l-2}, \mathcal{C}_{l-1}, \mathcal{C}_l$ are required to have the same spatial and channel dimensionalities. We instead allow the network to select different resolutions for each cell. To handle the divergence of resolutions in neighbouring cells, we first check their resolutions and adjust them accordingly by upsampling or downsampling if there is a mismatch.

**Residual Cell.** Previous studies opt for concatenating the outputs of all intermediate nodes to form the output of a cell (*e.g.,*[27, 18, 25]). We refer to such a design as a direct cell. Inspired by the residual connection in ResNet [28], we propose to also include the input of the cell in forming the output. See Figure 3 where the residual connection cells highlighted with a red line. This allows the network to learn *residual* mappings on top of direct mappings. Hereafter, we call this design *the residual cell*. We empirically find that residual cells outperform the direct ones (see § 3.3).

**Candidate Operation Selection.** The candidate operations for the feature net and matching net differ due to their functionalities. In particular, the feature net aims to extract distinctive local features for comparing pixel-wise similarities. We empirically observe that removing two commonly used operations in DARTS, namely **1.** *dilated separable convolutions* and **2.** *the pooling layers* does not hurt the performance. Thus, the set of candidates operators for the feature net includes $\mathcal{O}^F = \{$ "$3 \times 3$ 2D convolution", "skip connection"$\}$. Similarly, we find out that removing some of the commonly used operations from the candidate set for the matching net will not hurt the design. As such, we only include the following operations for the matching net, $\mathcal{O}^M = \{$ "$3 \times 3 \times 3$ 3D convolution", "skip connection"$\}$. We will shortly see an ablation study regarding this (see § 3.3).

### 2.1.2 Network Level Search Space

We define the network level search space as the arrangement of cells, which controls the variations in the feature dimensionality and information flow between cells. Drawing inspirations from [18], we aim to find an optimal path within a pre-defined $L$-layer trellis as shown in Figure 3. We associated a scalar with each gray arrow in that trellis. We use $\beta$ to represent the set of this scalar. Considering the number of filters in each cell, we follow the common practice of doubling the number when halving the height and width of the feature tensor.

In the network level search space, we have two hyper-parameters to set: **I.** the smallest spatial resolution and **II.** the number of layers $L$. Empirically, we observe that setting the smallest spatial resolution to $1/24$ of the input images work across a broad range of benchmarks and hence our choice in here. Based on this, we propose a four-level trellis with downsampling rates of $\{3, 2, 2, 2\}$, leading to the smallest feature map to be 1/24 of the input size (see Figure 3). In comparison to $\{2, 2, 2, 2, 2\}$, the downsampling to $1/3$ will remove the need of upsampling twice and we empirically observed similar performance. At the beginning of the feature net, we have a three-layer "stem" structure, the first layer of it is a $3 \times 3$ convolution layer with stride of three, followed by two layers of $3 \times 3$ convolution with stride of one.

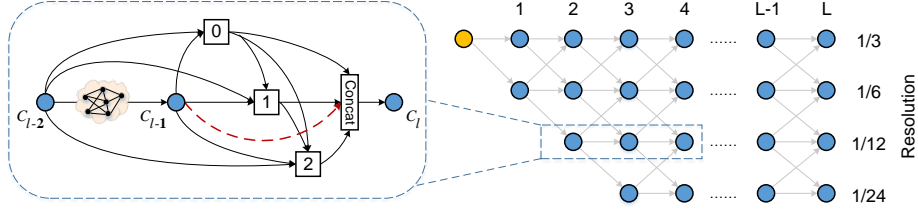

Figure 3: **The proposed search space.** We illustrate our cell level search space on the left and our network level search space on the right. The red dash line on the left represents the proposed residual connection. We set $L^F = 6$ for Feature Net and $L^M = 12$ for Matching Net.

Turning our attention to the the number of layers $L$, we have empirically observed that choosing $L^F = 6$ for the feature net and $L^M = 12$ for the matching net provides a good balance between the computational load and performance of the network. This is interestingly much smaller than some recent developments in hand-crafting deep stereo matching networks. For example, GA-Net [24] uses 33 convolutional layers with an hourglass structure to extract features.

Similar to finding the best operation between the nodes, we will use a set of search parameters $\boldsymbol{\beta}$ to search over the trellis in order to find a path in it that minimizes the loss. As shown in Fig 3, each cell in a level of the trellis can receive inputs from the preceding cell in the same level, one level below and one level above (if any of the latter two exists).

## 2.2   Loss Function and Optimization

Since our network can be searched and trained in an end-to-end manner, we directly apply supervisions on the output disparity maps, allowing the Feature Net and the Matching Net to be jointly searched. We use the smooth $\ell_1$ loss as our training loss function as it is considered to be robust to disparity discontinuities and outliers. Given the ground truth disparity $\mathbf{d}_{gt}$, the loss function is defined as:

$$\mathcal{L} = \ell(\mathbf{d}_{\mathrm{pred}} - \mathbf{d}_{\mathrm{gt}}), \ \ \text{where } \ell(x) = \left\{ \begin{array}{ll} 0.5x^2, & |x| < 1, \\ |x| - 0.5, & \text{otherwise.} \end{array} \right. \tag{3}$$

After continuous relaxation, we can optimize the weight $\mathbf{w}$ of the network and the architecture parameters $\boldsymbol{\alpha}, \boldsymbol{\beta}$ through bi-level optimization. We parameterize the cell structure and the network structure with $\boldsymbol{\alpha}$ and $\boldsymbol{\beta}$ respectively. To speed up the search process, we use the first-order approximation [18].

To avoid overfitting, we use two disjoint training sets train**I** and train**II** for $\mathbf{w}$ and $\boldsymbol{\alpha}, \boldsymbol{\beta}$ optimization respectively. We do alternating optimization for $\mathbf{w}$ and $\boldsymbol{\alpha}, \boldsymbol{\beta}$:

- Update network weights $\mathbf{w}$ by $\nabla_{\mathbf{w}}\mathcal{L}(\mathbf{w}, \boldsymbol{\alpha}, \boldsymbol{\beta})$ on train**I**
- Update architecture parameters $\boldsymbol{\alpha}, \boldsymbol{\beta}$ by $\nabla_{\boldsymbol{\alpha},\boldsymbol{\beta}}\mathcal{L}(\mathbf{w}, \boldsymbol{\alpha}, \boldsymbol{\beta})$ on train**II**

When the optimization convergence, we decode the discrete cell structures by retaining the top-2 strongest operations from all non-zero operations for each node, and the discrete network structures by finding a path with maximum probability [18].

## 3   Experiments

In this section, we adopt SceneFlow dataset [3] as the source dataset to analyze our architecture search outcome. We then conduct the architecture evaluation on KITTI 2012 [29], KITTI 2015 [30] and Middlebury 2014 [31] benchmarks by inheriting the searched architecture from SceneFlow dataset. In our ablation study, we analyze the effect of changing search space as well as different search strategies.

## 3.1   Architecture Search

We conduct full architecture search on SceneFlow dataset [3]. It contains 35,454 training and 4370 testing synthetic images with a typical image dimension of $540 \times 960$. We use the "finalpass" version as it is more realistic. We randomly select 20,000 image pairs from the training set as our search-training-set, and another 1,000 image pairs from the training set are used as the search-validation-set following [18].

**Implementation:**   We implement our LEAStereo network in Pytorch. Random crop with size $192 \times 384$ is the only data argumentation technique being used in this work. We search the architecture

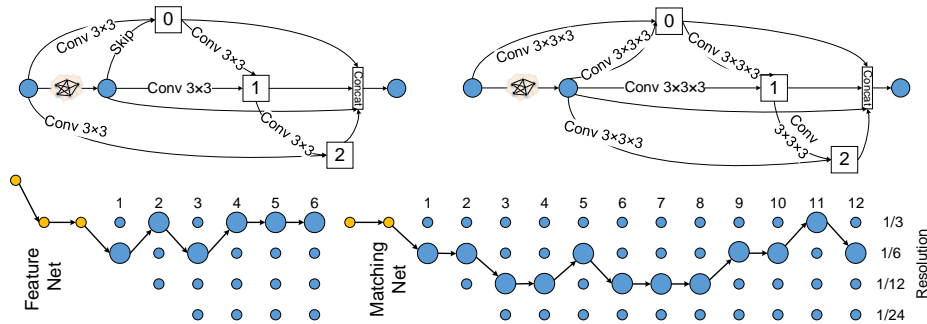

Figure 4: **The searched architecture.** The top two graphs are the searched cell structures for the Feature Net and Matching Net. The bottom are the searched network level structures for both networks. The yellow dots present the pre-defined "stem" layers and the blue dots denote the searchable cells.

for a total of 10 epochs: the first three epochs to initiate the weight **w** of the super-network and avoid bad local minima outcome; the rest epochs to update the architecture parameters $\alpha, \beta$. We use SGD optimizer with momentum 0.9, cosine learning rate that decays from 0.025 to 0.001, and weight decay 0.0003. The entire architecture search optimization takes about 10 GPU days on an NVIDIA V100 GPU.

The architecture found by our algorithm is shown in Figure 4. We manually add 2 skip connections for the Matching Net, one is between node 2 and 5, the other is 5 and 9. Noting that we only perform the architecture search once on the SceneFlow dataset and fine-tuned the searched architecture weights on each benchmark separately.

## 3.2 Benchmark Evaluation

**SceneFlow dataset**    We evaluate our LEAStereo network on SceneFlow [3] test set with 192 disparity level. We use average end point errors (EPE) and bad pixel ratio with 1 pixel threshold (bad 1.0) as our benchmark metrics. In Table 1, we can observe that LEAStereo achieves the best performance using only near one third of parameters in comparison to the SOTA hand-crafted methods (*e.g.,* [6]). Furthermore, the previous SOTA NAS-based method AutoDispNet [25] has 20× more parameters than our architecture. We show some of the qualitative results in Figure 7.

Table 1: **Quantitative results on Scene Flow dataset**. Our method achieves state-of-the-art performance with only a fraction of parameters. The parentheses indicate the test set is used for hyperparameters tuning.

| Methods | Params [M] | EPE [px] | bad 1.0 [%] | Runtime [s] |
|---|---|---|---|---|
| GCNet [4] | 3.5 | 1.84 | 15.6 | 0.9 |
| iResNet[5] | 43.34 | 2.45 | 9.28 | 0.2 |
| PSMNet [11] | 5.22 | 1.09 | 12.1 | 0.4 |
| GANet-deep [6] | 6.58 | 0.78 | 8.7 | 1.9 |
| AANet [26] | 3.9 | 0.87 | 9.3 | **0.07** |
| AutoDispNet [25] | 37 | (1.51) | - | 0.34 |
| LEAStereo | **1.81** | **0.78** | **7.82** | 0.3 |

**KITTI benchmarks**    As shown in Table 2 and the leader board, our LEAStereo achieves top 1 rank among other human designed architectures on KITTI 2012 and KITTI 2015 benchmarks. Figure 5 provides some visualizations from the testsets.

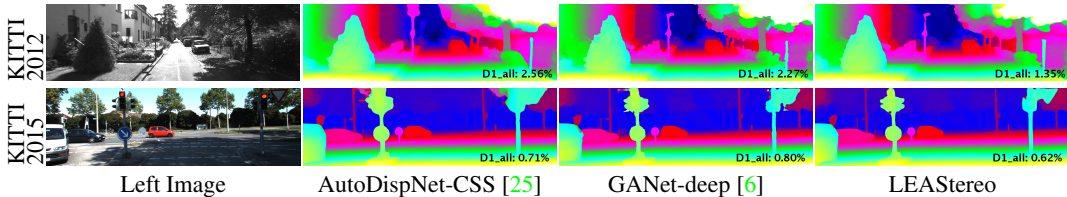

Figure 5: **Qualitative results on KITTI 2012 and KITTI 2015 benchmarks.**

**Middlebury 2014**    Middlebury 2014 is often considered to be the most challenging dataset for deep stereo networks due to restricted number of training samples and also the high resolution imagery

Table 2: **Quantitative results on the KITTI 2012 and 2015 benchmark.** Bold indicates the best.

| Methods | KITTI 2012 | | | | KITTI 2015 | | | |
| | bad 2.0 noc [%] | bad 3.0 noc [%] | Refl 3.0 noc [%] | Avg All [px] | FG Non Occ[%] | Avg All Non Occ[%] | FG All Areas[%] | Avg All All Areas[%] |
|---|---|---|---|---|---|---|---|---|
| GCNet [4] | 2.71 | 1.77 | 10.80 | 0.7 | 5.58 | 2.61 | 6.16 | 2.87 |
| PSMNet [11] | 2.44 | 1.49 | 8.36 | 0.6 | 4.31 | 2.14 | 4.62 | 2.32 |
| GANet-deep [6] | **1.89** | 1.19 | 6.22 | 0.5 | 3.39 | 1.84 | 3.91 | 2.03 |
| DispNetC [3] | 7.38 | 4.11 | 16.04 | 1.0 | 3.72 | 4.05 | 4.41 | 4.34 |
| AANet [26] | 2.90 | 1.91 | 10.51 | 0.6 | 1.80 | 2.32 | 1.99 | 2.55 |
| AutoDispNet-CSS [25] | 2.54 | 1.70 | 5.69 | 0.5 | 2.98 | 2.00 | 3.37 | 2.18 |
| LEAStereo | 1.90 | **1.13** | **5.35** | **0.5** | **2.65** | **1.51** | **2.91** | **1.65** |

with many thin objects. The full resolution of Middlebury is up to $3000 \times 2000$ with 800 disparity levels which is prohibitive for most deep stereo methods. This has forced several SOTA [11, 32] to operate on quarter-resolution images where details can be lost. In contrast, the compactness of our searched architecture compactness allows us to use images of size $1500 \times 1000$ with 432 disparity levels. In Table 3, we report the latest benchmark of volumetric based stereo matching method [11] and direct regression approaches [5, 26] for comparisons. Our proposed LEAStereo achieves the state-of-the-art rank on various metrics (*e.g.,*bad 4.0, all) among more than 120 stereo methods from the leader board. We also note that our network has better performance on large error thresholds, which might because of the downsampling operations prohibit the network to learn sub-pixel accuracy or the loss function that does not encourage sub-pixel accuracy.

Table 3: **Quantitative results on the Middlebury 2014 Benchmark.** Bold indicates the best. The red number on the top right of each number indicates the actual ranking on the benchmark.

| Methods | bad 2.0 [%] | | bad 4.0 [%] | | Ave Err [px] | | RMSE [px] | | A90 [px] | | A95 [px] | |
| | nonocc | all | nonocc | all | nonocc | all | nonocc | all | nonocc | all | nonocc | all |
|---|---|---|---|---|---|---|---|---|---|---|---|---|
| PSMNet [11] | 42.1[108] | 47.2[104] | 23.5[97] | 29.2[94] | 6.68[69] | 8.78[43] | 19.4[50] | 23.3[22] | 17.0[77] | 22.8[51] | 31.3[65] | 43.4[32] |
| iResNet [5] | 22.9[62] | 29.5[61] | 12.6[55] | 18.5[49] | 3.31[30] | 4.67[7] | 11.3[8] | 13.9[6] | 6.61[49] | 10.6[24] | 12.5[37] | 20.7[7] |
| AANet+ [26] | 15.4[44] | 22.0[40] | 10.8[45] | 16.4[42] | 6.37[66] | 9.77[49] | 23.5[73] | 29.4[42] | 7.55[56] | 29.3[58] | 48.8[80] | 76.1[63] |
| HSM [33] | 10.2[26] | 16.5[19] | 4.83[17] | 9.68[8] | 2.07[2] | 3.44[2] | 10.3[4] | 13.4[3] | 2.12[20] | 4.26[5] | 4.32[6] | 17.6[4] |
| LEAStereo | **7.15**[10] | **12.1**[5] | **2.75**[1] | **6.33**[1] | **1.43**[1] | **2.89**[1] | **8.11**[1] | **13.7**[5] | **1.68**[8] | **2.62**[1] | **2.65**[1] | **6.35**[1] |

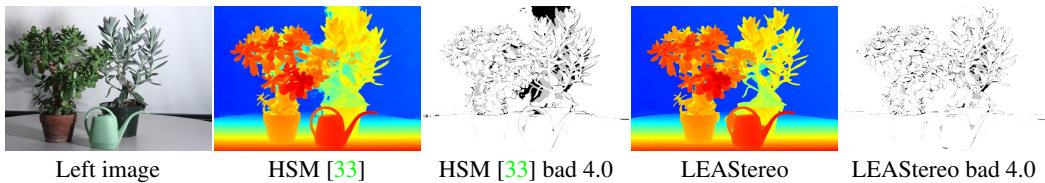

| Left image | HSM [33] | HSM [33] bad 4.0 | LEAStereo | LEAStereo bad 4.0 |

Figure 6: **Qualitative results on Middlebury 2014 Benchmark.**

### 3.3 Ablation Study

In this part, we perform ablation studies using the SceneFlow dataset [3] to justify "hyper-parameters" of our algorithm. In particular, we look into the candidate operations $\mathcal{O}$, differences between the residual and the direct cells, joint-search *vs.* separate-search of the Feature Net and the Matching Net, and functionality analysis for each sub-net. We also provide a head-to-head comparison to AutoDispNet [25].

**Candidate Operations**  Here we evaluate two sets of candidate operations $\mathcal{O}_{large}$ and $\mathcal{O}_{ours}$. The $\mathcal{O}_{large}$ consists of 8 operations including various types of convolution filters, pooling and connection mechanisms that are commonly used in [27, 18, 25][4]. The $\mathcal{O}_{ours}$ as described in § 2.1.1 only contains $3 \times 3$ convolution, skip connection and zero connection. For the Matching Net, we simply use the

Table 4: **Ablation Studies of different searching strategies.** The input resolution is $576 \times 960$, and EPE is measured on total SceneFlow test set.

| Architecture Variant | | | | | | Achieved Network | | |
|---|---|---|---|---|---|---|---|---|
| $\mathcal{O}_{large}$ | $\mathcal{O}_{ours}$ | Sepa. | Join. | Direct | Residual | Params | FLOPS | EPE |
| √ | | √ | | | √ | 0.68M | 682.8G | 1.55px |
| | √ | √ | | | √ | 2.00M | 782.4G | 0.86px |
| | √ | | √ | √ | | 1.52M | 538.9G | 0.91px |
| | √ | | √ | | √ | 1.81M | 782.2G | 0.78px |

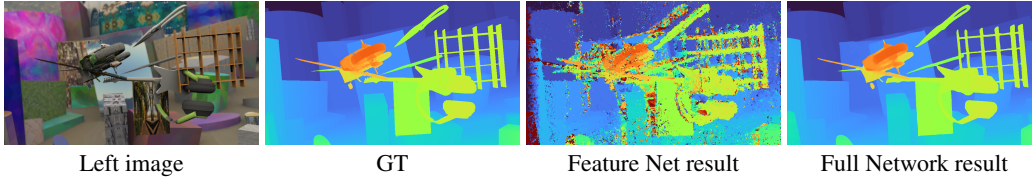

Left image        GT        Feature Net result        Full Network result

Figure 7: **Functionality analysis for the Feature Net and the Matching Net.** By using the learned features from the Feature Net directly, we already achieve reasonably good results as shown in the third sub-figure. After applying the Matching Net, we could further improve the results significantly as evidenced by the gap between the third and fourth sub-figures.

3D variants of these operations. As shown in the first two rows of Table 4, larger set of operations pool $\mathcal{O}_{large}$ leads to a searched architecture with a much lower number of parameters but poor EPE performance when compared to $\mathcal{O}_{ours}$. The reason is that the algorithm favors skip connections in its design, leading to a low-capacity architecture. Similar observations, albeit in another context, are reported in [34].

**Joint-search *vs.* Separate-search** To analyze the effectiveness and efficiency of joint-search vs. separate-search, we report their performances on SceneFlow dataset with $\mathcal{O}$ and connection type fixed. From the metrics of row 2 and 4 in Table 4, we observe that joint-search outperforms separate-search by an improved margin of $9.30\%$ for EPE and $10.50\%$ reduction on the number of parameters. We conjecture that the joint-search increases the capacity and compatibility for both Feature Net and Matching Net.

**Residual cell *vs.* Direct cell** We then study the differences between our proposed residual cell and direct cell. As shown in the last two rows of Table 4, using residual cell slightly generates more parameters and FLOPs but increase the performance by $14.29\%$.

**Functionality analysis for each sub-net** The Feature Net and the Matching Net own different roles in stereo matching. The Feature Net is designed to extract distinctive features from stereo pairs while the Matching Net is to compute matching costs from these features. To analyze and reflect the actual behaviour of each searched sub-net, we use the features from the Feature Net to directly compute a cost volume with dot products [2] and project it to a disparity map with the Winner-Takes-All (WTA) strategy. As shown in Figure 7, this strategy already achieves a reasonably good result in correctly estimating disparity for most objects, which demonstrates that our Feature Net is learning discriminative features for stereo matching. The difference between the third and fourth sub-figures (before and after the Matching Net) proves the contribution of the Matching Net in computing and aggregating the matching costs to achieve much better results.

### 3.4 Discussion

**LEAStereo *vs.* AutoDispNet** AutoDispNet [25] has a very different network design philosophy than ours. It is a large U-Net-like architecture and tries to directly regress disparity maps from input images in pixel space. In contrast, our design benefits from task-specific physics and inductive bias, *ie.,*the gold standard pipeline for deep stereo matching and the refine search space, thus achieves full architecture search within current physical constraints. Table 5 provides a head-to-head comparison on KITTI Benchmark. It is worth noting that our method is $32.12\%$ better than AutoDispNet-CSS [25] with only $1.81$M parameters (which is $1.7\%$ of the parameters required by AutoDispNet-CSS!)

**Hints from the found architecture** There are several hints from the found architecture: 1. The feature net does not need to be too deep to achieve a good performance; 2. Larger feature volumes lead to better performance ($1/3$ is better than $1/6$); 3. A cost volume of $1/6$ resolution seems

Table 5: **LEAStereo** *vs.* **AutoDispNet**

| Methods | Search Level | Params | KITTI 2012 | KITTI 2015 | Runtime |
|---|---|---|---|---|---|
| AutoDispNet-CSS | Cell | 111M | 1.70% | 2.18% | 0.9 s |
| LEAStereo | Full Network | 1.8M | 1.13% | 1.65% | 0.3 s |

proper for good performance; 4. Multi-scale fusion seems important for computing matching costs (*ie.,*using a DAG to fuse multi-scale information). Note that the fourth hint has also been exploited by AANet [26] which builds multiple multi-scale cost volumes and regularizes them with a multi-scale fusion scheme. Our design is tailored for the task of stereo matching and may not be considered as a domain agnostic solution for a different problem. For different domains, better task-dependent NAS mechanisms are still a suitable solution to efficiently incorporate inductive bias and physics of the problem into the search space.

## 4   Related Work

**Deep Stereo Matching**   MC-CNN [2] is the first deep learning based stereo matching method. It replaces handcrafted features with learned features and achieves better performance. DispNet [3] is the first end-to-end deep stereo matching approach. It tries to directly regress the disparity maps from stereo pairs. The overall architecture is a large U-shape encoder-decoder network with skip connections. Since it does not leverage on pre-acquired human knowledge in stereo matching, this network is totally data-driven, and requires large training data and often hard to train. GC-Net [4] used a 4D feature volume to mimic the first step of conventional stereo matching pipeline and a soft-argmin process to mimic the second step. By encoding such human knowledge in network design, training becomes easier while maintaining high accuracies. Similar to our work, GC-Net also consists of two sub-networks to predict disparities. GA-Net [33] proposes a semi-global aggregation layer and a local guided aggregation layer to capture the local and the whole-image cost dependencies respectively. Generally speaking and as alluded to earlier, designing a good structure for stereo matching is very difficult, despite considerable effort put in by the vision community.

**Neural Architecture Search for Dense Predictions**   Rapid progress on NAS for image classification or object detection has been witnessed as of late. In contrast, only a handful of studies target the problem of dense predictions such as scene parsing, and semantic segmentation. Pioneer works [35, 36] propose a super-net that embed a large number of architectures in a grid arrangement and adopt them for the task of semantic segmentation. To deal with the explosion of computational demands in dense prediction, Chen *et al.* [37] employ a handcrafted backbone and only search the decoder architecture. Rather than directly searching architectures on large-scale dense prediction tasks, they design a small scale proxy task to evaluate the searching results. Nekrasov *et al.* [38] focus on the compactness of a network and utilized a small-scale network backbone with over-parameterised auxiliary searchable cells on top of it. Similarly, Zhang *et al.*[19] penalized the computational demands in search operations, allowing the network to search an optimized architecture with customized constraints. Auto-Deeplab [18] proposes a hierarchical search space for semantic segmentation, allowing the network to self-select spatial resolution for encoders. FasterSeg [20] leverages on the idea of [19] and [18], and introduces multi-resolution branches to the search space to identify an effective semantic segmentation network. AutoDispNet [25] applies NAS to disparity estimation by searching cell structures for a large-scale U-shape encoder-decoder structure.

## 5   Conclusion

In this paper, we proposed the first end-to-end hierarchical NAS framework for deep stereo matching, which incorporates task-specific human knowledge into the architecture search framework. Our search framework tailors the search space and follows the feature net-feature volume-matching net pipeline, whereas we could optimize the architecture of the entire pipeline jointly. Our searched network outperforms all state-of-the-art deep stereo matching architectures (handcrafted and NAS searched) and is ranked at the top 1 accuracy on KITTI stereo 2012, 2015 and Middlebury benchmarks while showing substantial improvement on the network size and inference speed. In the future, we plan to extend our search framework to other dense matching tasks such as optical flow estimation [39] and multi-view stereo [40].

## Broader Impact

The task of stereo matching has been studied for over half a century and numerous methods have been proposed. From traditional methods to deep learning based methods, people keep setting a new state-of-the-art through these years. Nowadays, deep learning based methods become more popular than traditional methods since deep methods are more accurate and faster. However, finding a better architecture for stereo matching networks remains a hot topic recently. Rather than designing a handcrafted architecture with trial and error, we propose to allow the network to learn a good architecture by itself in an end-to-end manner. Our method reduces more than $2/3$ of searching time than previous method [25] and has much better performance, thus saves lots of energy consumption and good for our planet by reducing massive carbon footprints.

In addition, our proposed search framework is relatively general and not limited to the specific task of stereo matching. It can be well extended to other dense matching tasks such as optical flow estimation and multi-view stereo.

## Acknowledgments and Disclosure of Funding

Yuchao Dai's research was supported in part by Natural Science Foundation of China (61871325, 61671387) and National Key Research and Development Program of China under Grant 2018AAA0102803. Hongdong Li's research was supported in part by the ARC Centre of Excellence for Robotics Vision (CE140100016) and ARC-Discovery (DP 190102261), ARC-LIEF (190100080) grants. Zongyuan Ge and Xuelian Cheng were supported by Airdoc Research Australia Centre Funding. Zongyuan Ge was also supported by Monash-NVIDIA joint Research Centre.

## Footnotes

* Indicates equal contribution

[1]To the best of our knowledge.

[2]At the time of submitting this draft, our algorithm, LEAStereo, is ranked **1** in the KITTI 2015, and KITTI 2012 benchmark, and ranked **1** according to Bad 4.0, avgerr, rms, A95, A99 metrics on Middlebury benchmark.

[3]We would like to stress that our framework, in contrast to the Auto-DeepLab, searches for the *full* architecture (Auto-DeepLab only searches for the architecture of the encoder).

[4]Namely, $\mathcal{O}_{large}$ contains $3 \times 3$ and $5 \times 5$ depth-wise separable convolutions, $3 \times 3$ and $3 \times 3$ dilated separable convolutions with dilation factor 2, skip connection, $3 \times 3$ average pooling, $3 \times 3$ max pooling and zero connection.

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
