[Supplementary Material]

# Hierarchical Neural Architecture Search for Deep Stereo Matching – Supplementary Materials

*Xuelian Cheng[1,5], *Yiran Zhong[2,6], Mehrtash Harandi[1,7], Yuchao Dai[3], Xiaojun Chang[1],
Tom Drummond[1], Hongdong Li[2,6], Zongyuan Ge[1,4,5]

[1]Faculty of Engineering, Monash University, [2]Australian National University,
[3]Northwestern Polytechnical University, [4]eResearch Centre, Monash University,
[5]Airdoc Research Australia, [6]ACRV, [7]Data61, CSIRO

{xuelian.cheng,mehrtash.harandi}@monash.edu,
{xiaojun.chang,tom.drummond,zongyuan.ge}@monash.edu,
{yiran.zhong, hongdong.li}@anu.edu.au, daiyuchao@nwpu.edu.cn

In this supplemental material, we briefly introduce three widely used stereo matching benchmarks, provide details of the separate-search (§ 3.3 of the main manuscript) of the Feature Net and the Matching Net, and show more qualitative results of our method on various datasets and screenshots of benchmarks.

## 1 Benchmark Datasets

**KITTI 2012 and 2015 datasets**    These two datasets are both real-world datasets collected from a driving car. KITTI 2012 contains 194 training image pairs and 195 test image pairs. KITTI 2015 contains 200 stereo pairs for training and 200 for testing. The typical resolution of KITTI images is $376 \times 1240$. For KITTI 2012, the semi-dense ground truth disparity maps are generated by Velodyne HDL64E LiDARs, while for KITTI 2015, 3D CAD models for cars are manually inserted [1]. From the training set, we use 180 images for training and 20 images for validation for KITTI 2015. For KITTI 2012, we use 174 images from the training set for training and 20 from the training set for validation. We use a maximum disparity level of 192 in this dataset. For both datasets, the ground truth of test set benchmarks are withheld from the participating algorithms and all evaluations are done online (where participants submit their results to be evaluated).

**Middlebury 2014 dataset**    Middlebury 2014 contains 15 images for training and 15 images for testing. Most of the stereo pairs are indoor scenes with handcrafted layouts. As a real-world dataset, the ground truth disparities are captured by structured light with high density and at sub-pixel accuracy. This dataset contains many thin objects and large disparity ranges. The full resolution of Middlebury dataset is up to $3000 \times 2000$ with 800 disparity levels. Since it uses a bad-pixel-ratio with a threshold of 0.5 pixels as a benchmark metric, it encourages the algorithms to have sub-pixel accuracy. Similar to the KITTI datasets, the ground truth of the test set are withheld in the online benchmark.

## 2 Details of Separate-search

The Feature Net and the Matching net are closely connected: without the Matching Net, we could not generate the final disparity map and the Matching Net depends on the features from the Feature Net. In the separate-search scheme, we first search a Feature Net structure using our proposed feature loss and then search the Matching Net structure along with the searched Feature Net. The proposed feature loss provides a direct supervision on the Feature Net, allowing us to search it without the Matching Net.

Let $\mathbf{f}^L, \mathbf{f}^R : \mathbb{R}^{3 \times H \times W} \to \mathbb{R}^{c \times H \times W}$ be the $c$-dimensional feature maps of the left and right images with a resolution of $H \times W$. We construct a 3D cost volume by computing the inner product between

each pixel in the left feature map $\mathbf{f}^L$ with a set of disparity shifted right feature map $\mathbf{f}^R$. For pixel $\mathbf{x}$ with a given disparity shift $d \in \{0, 1, 2, \cdots, D\}$, the matching cost is computed as:

$$C(\mathbf{x}, d) = < \mathbf{f}^L(\mathbf{x}), \mathbf{f}^R(\mathbf{x} - d) >, \quad C \in \mathbb{R}^{D \times H \times W}, \tag{1}$$

and the matching probability volume can be computed by applying a softmax operation to the cost volume:

$$\mathbf{P}_{\mathrm{pred}}(\mathbf{p}, d) = \mathrm{softmax}\left(-C(\mathbf{p}, d)\right), \tag{2}$$

The target of the Feature Net is to extract distinctive features, which means the matching probability for each pixel should be unimodal rather than mutlimodal. To achieve this goal, we propose a direct supervision on $\mathbf{P}_{\mathrm{pred}}$.

We first generate a ground truth unimodal matching probability volume $\mathbf{P}_{\mathrm{gt}}$: for each pixel, we create a Laplace distribution with $\mu = d_{\mathrm{gt}}, b = 0.01$, where $\mu$ is a location parameter, $b$ is the diversity and $d_{\mathrm{gt}}$ is ground truth disparity. Therefore, we have two distributions, $\mathbf{P}_{\mathrm{pred}}, \mathbf{P}_{\mathrm{gt}}$. To evaluate the differences between these two distributions, empirically we observed that the $\ell_2$ loss works better in comparison with popular choices like cross-entropy or focal loss [2] in our case.

Given the matching probability volume, we can also generate a disparity map using soft-argmin operation $\mathbf{d}_{\mathrm{feature}} = \sum_{d=0}^{D}(d \times \mathbf{P}_{\mathrm{pred}}(d))$. Therefore, we add another disparity supervision in our feature loss. Our overall feature loss is then written as:

$$\mathcal{L}_{\mathrm{feature}} = \|\mathbf{d}_{\mathrm{feature}} - \mathbf{d}_{\mathrm{gt}}\|_1 + \lambda \|\mathbf{P}_{\mathrm{pred}} - \mathbf{P}_{\mathrm{gt}}\|_2^2, \tag{3}$$

where $\mathbf{d}_{\mathrm{gt}}$ is the ground truth disparity map, and $\lambda = 0.01$.

## 3 Quantitative Results

We provide more qualitative results on the SceneFlow, KITTI 2012, KITTI 2015 and Middlebury datasets in Figure 1 2 3 4, respectively. We notice that our method can successfully recover sharp boundaries and thin objects structures.

| Left image | GT | LEAStereo | Error map |

Figure 1: **Qualitative comparison on the SceneFlow dataset.**

## 4 Screenshots of KITTI and Middlebury Public Table

In Figure 5 6, we show the screenshots of LEAStereo on KITTI 2012, KITTI 2015 and Middlebury 2014 public tables respectively. Our method is ranked **1** on all benchmarks as of May 29th, 2020.

| Left image | LEAStereo | Error map |

Figure 2: **Qualitative results on KITTI 2012.**

| Left image | LEAStereo | Error map |

Figure 3: **Qualitative results on KITTI 2015.**

| Left image | LEAStereo | Left image | LEAStereo |

Figure 4: **Qualitative results on Middlebury 2014 Benchmark.**

## KITTI 2012 Stereo benchmark

Table: [All ▼]    Error threshold: [3 pixels ▼]    Evaluation area: [All pixels ▼]

| | Method | Setting | Code | Out-Noc | Out-All | Avg-Noc | Avg-All | Density | Runtime | Environment | Compare |
|---|---|---|---|---|---|---|---|---|---|---|---|
| 1 | FDCVNet | | | 0.00 % | 0.00 % | 0.0 px | 0.0 px | 0.00 % | 0.03 s | 1 core @ 2.5 Ghz (C/C++) | ☐ |
| 2 | attention_global_net | | | 0.00 % | 0.00 % | 0.0 px | 0.0 px | 0.00 % | 1.5 s | 4 cores @ 2.5 Ghz (Python) | ☐ |
| | ERROR: Wrong syntax in BIBTEX file. | | | | | | | | | | |
| 3 | | | | 0.00 % | 0.00 % | 0.0 px | 0.0 px | 0.00 % | | | ☐ |
| 4 | LEAStereo | | | 1.13 % | 1.45 % | 0.5 px | 0.5 px | 100.00 % | 0.3 s | GPU @ 2.5 Ghz (Python) | ☐ |
| 5 | MSMD-Net | | | 1.14 % | 1.47 % | 0.4 px | 0.5 px | 100.00 % | 0.72 s | 1 core @ 2.5 Ghz (C/C++) | ☐ |

## KITTI 2015 Stereo benchmark

Evaluation ground truth: [All pixels ▼]    Evaluation area: [All pixels ▼]

| | Method | Setting | Code | D1-bg | D1-fg | D1-all | Density | Runtime | Environment | Compare |
|---|---|---|---|---|---|---|---|---|---|---|
| 1 | LEAStereo | | | 1.40 % | 2.91 % | 1.65 % | 100.00 % | 0.30 s | GPU @ 2.5 Ghz (Python) | ☐ |
| 2 | Dahua_Stereo | | | 1.48 % | 2.83 % | 1.71 % | 100.00 % | 1.52 s | GPU @ 2.5 Ghz (Python) | ☐ |
| 3 | MSMD-Net(only MS) | | | 1.41 % | 3.22 % | 1.71 % | 100.00 % | 0.52 s | 1 core @ 2.5 Ghz (C/C++) | ☐ |
| 4 | CSPN | | | 1.51 % | 2.88 % | 1.74 % | 100.00 % | 1.0 s | GPU @ 2.5 Ghz (Python) | ☐ |

X. Cheng, P. Wang and R. Yang: Learning Depth with Convolutional Spatial Propagation Network. IEEE Transactions on Pattern Analysis and Machine Intelligence(T-PAMI) 2019.

Figure 5: **Screenshot of the KITTI 2012 and 2015 public table as of May 29th, 2020.**

---

Set: **test dense** test sparse training dense training sparse
Metric: **bad 0.5 bad 1.0 bad 2.0 bad 4.0** avgerr rms A50 A90 A95 A99 time time/MP time/GD
Mask: **nonocc** all
☐ plot selected ☐ show invalid Reset sort Reference list

| Date | Name | Res | Avg (bad 4.0 %) | Austr | AustrP | Bicyc2 | Class | ClassE | Compu | Crusa | CrusaP | Djemb | DjembL | Hoops | Livgrm | Nkuba | Plants | Stairs |
|---|---|---|---|---|---|---|---|---|---|---|---|---|---|---|---|---|---|---|
| 05/28/20 ☑ | LEAStereo | H | 2.75 (1) | 3.57 (14) | 2.46 (6) | 1.68 (3) | 1.60 (6) | 3.47 (4) | 1.87 (5) | 1.57 (1) | 1.59 (5) | 1.44 (3) | 4.02 (3) | 4.11 (1) | 5.01 (3) | 5.13 (1) | 2.89 (1) | 3.22 (1) |
| 05/26/18 ☐ | NOSS_ROB | H | 3.46 (2) | 2.69 (4) | 2.37 (4) | 1.99 (7) | 1.19 (2) | 3.11 (1) | 1.75 (3) | 1.63 (5) | 1.26 (1) | 1.87 (15) | 5.60 (5) | 4.63 (3) | 5.66 (4) | 10.4 (19) | 3.60 (9) | 7.02 (22) |
| 03/09/19 ☐ | 3DMST-CM | H | 3.57 (3) | 3.07 (10) | 2.77 (11) | 1.80 (4) | 1.79 (10) | 5.43 (16) | 2.58 (10) | 1.71 (7) | 2.01 (15) | 1.48 (4) | 6.86 (9) | 5.14 (5) | 4.41 (1) | 8.85 (8) | 3.84 (12) | 6.29 (12) |

Set: **test dense** test sparse training dense training sparse
Metric: bad 0.5 bad 1.0 bad 2.0 **bad 4.0** avgerr rms A50 A90 A95 A99 time time/MP time/GD
Mask: nonocc **all**
☐ plot selected ☐ show invalid Reset sort Reference list

| Date | Name | Res | Avg (bad 4.0 %) | Austr | AustrP | Bicyc2 | Class | ClassE | Compu | Crusa | CrusaP | Djemb | DjembL | Hoops | Livgrm | Nkuba | Plants | Stairs |
|---|---|---|---|---|---|---|---|---|---|---|---|---|---|---|---|---|---|---|
| 05/28/20 ☑ | LEAStereo | H | 6.33 (1) | 4.59 (7) | 3.55 (2) | 3.16 (1) | 5.63 (1) | 7.77 (1) | 8.46 (1) | 5.86 (4) | 4.77 (1) | 2.97 (1) | 5.94 (3) | 13.6 (1) | 9.11 (4) | 9.13 (1) | 7.14 (1) | 6.83 (1) |
| 03/09/19 ☐ | 3DMST-CM | H | 8.10 (2) | 4.66 (8) | 4.36 (10) | 4.35 (2) | 6.89 (5) | 11.2 (9) | 11.5 (6) | 5.51 (2) | 6.09 (4) | 3.35 (3) | 8.77 (9) | 14.8 (4) | 6.93 (1) | 15.2 (9) | 9.87 (5) | 15.1 (16) |
| 05/26/18 ☐ | NOSS_ROB | H | 8.37 (3) | 3.96 (2) | 3.62 (4) | 5.01 (6) | 6.62 (2) | 8.72 (5) | 10.6 (3) | 7.53 (10) | 8.34 (13) | 3.60 (9) | 7.71 (5) | 14.5 (3) | 8.42 (3) | 16.5 (14) | 10.2 (6) | 13.5 (9) |

Figure 6: **Screenshot of the Middlebury Stereo Evaluation - Version 3 as of May 29th, 2020.**

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

☐ plot selected  ☐ show invalid   Reset sort   Reference list

| Date | Name | Res | avgerr (pixels) Avg | Austr | AustrP | Bicyc2 | Class | ClassE | Compu | Crusa | CrusaP | Djemb | DjembL | Hoops | Livgrm | Nkuba | Plants | Stairs |
|---|---|---|---|---|---|---|---|---|---|---|---|---|---|---|---|---|---|---|
| 05/28/20 ✅ LEAStereo | | H | **1.43** 1 | 2.18 1 | 1.90 1 | 1.11 3 | 1.11 3 | 1.27 1 | 1.15 2 | 1.31 1 | 1.30 3 | 0.71 11 | 1.28 4 | 1.84 1 | 1.74 1 | 1.64 1 | 1.89 1 | 1.47 1 |
| 11/08/18 ☐ HSM | | F | 2.07 2 | 2.89 17 | 2.26 6 | 1.74 25 | 1.22 6 | 2.46 10 | 1.44 15 | 1.46 3 | 1.27 2 | 0.70 9 | 2.57 19 | 3.34 19 | 2.16 4 | 2.21 3 | 4.75 30 | 2.00 2 |
| 05/26/18 ☐ NOSS_ROB | | H | 2.08 3 | 2.61 13 | 2.34 10 | 1.07 2 | 1.06 2 | 1.91 2 | 1.21 6 | 1.80 9 | 1.37 5 | 0.79 23 | 1.46 5 | 2.86 7 | 2.45 9 | 3.66 22 | 3.42 14 | 4.83 22 |

☐ plot selected  ☐ show invalid   Reset sort   Reference list

| Date | Name | Res | avgerr (pixels) Avg | Austr | AustrP | Bicyc2 | Class | ClassE | Compu | Crusa | CrusaP | Djemb | DjembL | Hoops | Livgrm | Nkuba | Plants | Stairs |
|---|---|---|---|---|---|---|---|---|---|---|---|---|---|---|---|---|---|---|
| 05/28/20 ✅ LEAStereo | | H | **2.89** 1 | 2.81 1 | 2.52 1 | 1.83 1 | 2.46 1 | 2.75 1 | 3.81 10 | 2.91 3 | 3.09 4 | 1.07 3 | 1.67 4 | 5.34 3 | 2.59 1 | 3.09 1 | 5.13 2 | 2.79 3 |
| 11/08/18 ☐ HSM | | F | 3.44 2 | 3.65 11 | 3.03 5 | 2.08 2 | 2.67 2 | 3.98 4 | 3.68 9 | 2.58 2 | 2.43 1 | 1.03 2 | 2.92 15 | 5.19 2 | 3.54 8 | 3.39 3 | 9.34 11 | 2.75 2 |
| 11/14/19 ☐ HSM-Smooth-Occ | | F | 3.44 3 | 3.55 7 | 3.09 6 | 3.56 34 | 2.93 3 | 3.91 3 | 3.46 7 | 2.41 1 | 2.43 1 | 1.11 4 | 2.82 13 | 6.26 5 | 3.32 4 | 3.24 2 | 7.59 3 | 3.28 4 |

☐ plot selected  ☐ show invalid   Reset sort   Reference list

| Date | Name | Res | A90 (pixels) Avg | Austr | AustrP | Bicyc2 | Class | ClassE | Compu | Crusa | CrusaP | Djemb | DjembL | Hoops | Livgrm | Nkuba | Plants | Stairs |
|---|---|---|---|---|---|---|---|---|---|---|---|---|---|---|---|---|---|---|
| 05/28/20 ✅ LEAStereo | | H | **2.62** 1 | 1.82 19 | 1.34 17 | 1.48 6 | 1.97 7 | 2.92 3 | 4.00 1 | 2.42 8 | 2.19 4 | 1.38 23 | 2.73 6 | 6.19 2 | 3.55 5 | 3.66 1 | 2.53 2 | 2.83 2 |
| 03/10/17 ☐ MC-CNN+TDSR | | F | 3.73 2 | 1.61 16 | 1.36 18 | 1.91 14 | 2.01 8 | 16.2 28 | 9.00 16 | 2.13 6 | 2.41 5 | 1.16 12 | 2.94 7 | 5.98 1 | 4.19 6 | 4.68 4 | 3.19 4 | 2.40 1 |
| 03/09/19 ☐ 3DMST-CM | | H | 3.95 3 | 1.24 9 | 1.13 11 | 1.26 2 | 1.76 5 | 7.33 14 | 7.00 8 | 1.74 1 | 1.82 1 | 1.20 14 | 3.13 8 | 11.5 7 | 2.56 2 | 11.6 21 | 3.73 5 | 8.12 15 |

☐ plot selected  ☐ show invalid   Reset sort   Reference list

| Date | Name | Res | A95 (pixels) Avg | Austr | AustrP | Bicyc2 | Class | ClassE | Compu | Crusa | CrusaP | Djemb | DjembL | Hoops | Livgrm | Nkuba | Plants | Stairs |
|---|---|---|---|---|---|---|---|---|---|---|---|---|---|---|---|---|---|---|
| 05/28/20 ✅ LEAStereo | | H | **2.65** 1 | 2.84 26 | 1.83 25 | 1.91 7 | 1.92 24 | 3.03 8 | 3.00 9 | 2.15 17 | 2.17 19 | 1.60 30 | 3.49 4 | 3.38 1 | 4.01 3 | 4.07 1 | 2.56 15 | 3.02 1 |
| 03/09/19 ☐ 3DMST-CM | | H | 3.23 2 | 1.57 9 | 1.31 9 | 1.38 2 | 1.43 14 | 5.37 16 | 2.00 1 | 1.58 3 | 1.70 7 | 1.26 18 | 6.80 9 | 4.17 5 | 3.11 1 | 11.7 27 | 2.32 10 | 7.40 18 |
| 01/24/17 ☐ 3DMST | | H | 3.47 3 | 1.50 8 | 1.21 3 | 1.89 6 | 1.39 10 | 4.00 14 | 2.00 1 | 1.60 6 | 1.70 7 | 1.24 14 | 11.1 18 | 6.36 16 | 5.87 5 | 10.2 14 | 2.15 5 | 5.32 9 |

☐ plot selected  ☐ show invalid   Reset sort   Reference list

| Date | Name | Res | A95 (pixels) Avg | Austr | AustrP | Bicyc2 | Class | ClassE | Compu | Crusa | CrusaP | Djemb | DjembL | Hoops | Livgrm | Nkuba | Plants | Stairs |
|---|---|---|---|---|---|---|---|---|---|---|---|---|---|---|---|---|---|---|
| 05/28/20 ✅ LEAStereo | | H | **6.35** 1 | 3.59 9 | 2.49 8 | 2.60 1 | 4.95 1 | 8.25 1 | 10.0 1 | 4.70 2 | 3.81 1 | 2.30 9 | 4.61 3 | 22.0 3 | 9.95 1 | 7.84 1 | 8.72 1 | 5.66 1 |
| 07/26/19 ☐ EdgeStereo | | F | 15.1 2 | 12.7 26 | 7.93 29 | 10.9 28 | 21.2 15 | 17.8 3 | 14.0 3 | 9.95 16 | 9.64 15 | 4.44 29 | 5.49 4 | 23.6 4 | 18.8 6 | 15.7 5 | 23.3 2 | 46.8 22 |
| 11/14/19 ☐ HSM-Smooth-Occ | | F | 16.3 3 | 5.79 13 | 4.03 13 | 12.5 33 | 12.2 5 | 22.2 5 | 16.0 7 | 6.17 7 | 7.76 8 | 2.93 17 | 13.6 14 | 27.4 6 | 19.5 10 | 11.1 3 | 71.5 9 | 11.6 5 |

☐ plot selected  ☐ show invalid   Reset sort   Reference list

| Date | Name | Res | A99 (pixels) Avg | Austr | AustrP | Bicyc2 | Class | ClassE | Compu | Crusa | CrusaP | Djemb | DjembL | Hoops | Livgrm | Nkuba | Plants | Stairs |
|---|---|---|---|---|---|---|---|---|---|---|---|---|---|---|---|---|---|---|
| 05/28/20 ✅ LEAStereo | | H | **20.2** 1 | 59.8 2 | 56.5 2 | 13.7 1 | 5.92 1 | 13.3 1 | 9.00 1 | 6.64 1 | 5.73 1 | 6.26 2 | 11.8 3 | 24.3 1 | 14.9 1 | 25.8 1 | 11.6 2 | |
| 11/08/18 ☐ HSM | | F | 39.2 2 | 78.5 7 | 72.7 9 | 42.8 20 | 10.7 8 | 38.2 11 | 13.0 3 | 7.69 2 | 6.88 3 | 6.20 1 | 32.9 13 | 95.5 18 | 46.2 4 | 20.0 2 | 126 17 | 32.3 3 |
| 07/26/19 ☐ EdgeStereo | | F | 40.8 3 | 62.8 3 | 60.3 3 | 33.9 11 | 32.1 28 | 25.3 5 | 14.0 5 | 48.3 40 | 43.8 40 | 19.4 55 | 22.0 4 | 84.9 12 | 43.9 1 | 28.6 10 | 53.5 2 | 69.7 6 |