[Reviews · NeurIPS 2020]

Review 1

Summary and Contributions: The paper presents an approach based on Neural Architecture Search (NAS) for optimizing a 3D convolutional network for stereo matching. While NAS has been successful in classification and other vision problems, it has not been implemented effectively in scenarios requiring per-pixel estimates, such as stereo matching. The authors argue that this is due to the extremely large search space in such problems. They propose to rely on domain expertise to guide the search reducing the search space. While the theoretical contribution is limited, the search space reduction allows NAS to converge on network configurations that achieve outstanding statistics on benchmarks, surpassing AutoDispNet [24], the previous attempt to apply NAS to stereo matching. The proposed LEAStereo achieves remarkable performance on four datasets. In fact, it ranks first on the two KITTI benchmarks and first according to a few criteria on the Middlebury benchmark which has been notoriously hard for end-to-end methods. (I find not testing the algorithm on the ETH3D benchmark acceptable.) I do have some reservations on the theoretical contribution of the method, but overall, I am leaning towards acceptance. The proposed approach strikes a balance between prior knowledge/heuristics and optimization and is able to find extremely effective architectures. My view would be even more favorable if it is clarified that the exact same architecture (with different weights) is used on all datasets. (See S4 below.)

Strengths: S1. The largest strength of the paper is the outstanding performance on the benchmarks, already discussed above. LEAStereo achieves remarkable rankings for an end-to-end method on the Middlebury benchmark, especially according to criteria that penalize gross outliers, such as average error, RMS, and bad 4.0. This benchmark has posed great challenges to end-to-end methods. See below for a related minor weakness. S2. NAS is performed hierarchically at both cell and network level structures. This goes beyond AutoDispNet [24], which is the previous state of the art in NAS for stereo matching. Similar strategies, however, have been applied to semantic segmentation. (Footnote 3 is not a strong argument.) S3. I consider the fact that the resulting networks are small compared to alternatives encouraging. It indicates that the proposed strategy allocates parameters to network components efficiently. S4. Lines 186-188 state that the same architecture is used on all datasets. This is not mentioned again, but if it is indeed the case, it means that the automatically discovered architecture has good generalization properties. More discussion is definitely needed, since it is not clear what enables generalization across datasets.

Weaknesses: W1. Reducing the set of candidate operations based on user intuition may lead to good performance in a given domain but does not shed light on how to apply the approach in different domains without comparable levels of expertise. The contribution of this paper is a very effective network for a given task, not an approach for creating effective networks for various tasks. W2 (minor). There are claims in the abstract and conclusions that the proposed approach improves the interpretability of the proposed network. This is not support sufficiently. Following this argument, the interpretability of networks generated by the proposed approach has to be inferior to the interpretability of CG-Net, for example, where each part of the network has been designed to match a part of the hand-crafted pipeline. It may be better than other NAS approaches, but this is not a strength of the method. W3. As mentioned above (S1), I consider performance on the Middlebury benchmark a strength. LEAStereo’s rankings are substantially worse for small error thresholds. Bad 1.0 and 2.0 errors are much worse than bad 4.0. I assume that this due to the downsampling operations in the network that allow it to regress a roughly accurate disparity map, but prevent these estimates from being very precise. This behavior should be discussed in the text.

Correctness: The claims are correct. I am not fully convinced by the interpretability claim (see above), but it is not a major flaw.

Clarity: Writing can be improved, but it is of sufficient quality.

Relation to Prior Work: Yes. Also, no critical references are missing.

Reproducibility: Yes

Additional Feedback: There are minor language errors throughout that can be corrected after careful proof-reading. They include number mismatches, missing or spurious articles, The chosen wording is often too informal. Minor comments (non-exhaustive list): 36-37: volumetric methods do not necessarily employ SGM. GC-Net and PSMNet are arguably the most popular volumetric methods and they do not use SGM. Also, 3D convolutional is probably a better term. Volumetric may suggest multi-view methods estimating occupancy probabilities. Footnote 2: “ranked first in” 88 and elsewhere: “explosion demands” is just wrong. “Explosive” is only slightly better. 99: “convergence” “performance” Table 2: “Ref” for reflective is confusing. “Refl” would have been better. ===== Post rebuttal comments ===== My concern about the heuristics needed to achieve good ranking in the benchmarks has not been addressed. It is an important aspect of the paper. There was an attempt in the rebuttal to address my comments on interpretability and low performance at low tolerance to errors. These are minor issues with the paper, that will be presented more clearly in the camera-ready version. The fact that the same architecture is applied to all datasets is positive. I am less positive than other reviewers because I find that approach overly reliant on heuristics, aiming to optimize its ranking in the benchmark at the expense of generality. I agree with them, however, that the paper is worthy of being presented at the conference.


Review 2

Summary and Contributions: This work introduces a hierarchical neural architecture search (NAS) for stereo matching. In [24], the NAS was applied to find an optimal architecture in the regression based stereo matching, but the performance is rather limited due to the inherent limitation of the direct regression in the stereo matching. Instead, this work automatically selects the optimal architecture for volumetric stereo matching consisting of feature net and matching net. Additionally, it searches for both the cell level and network level structures. These newly added components enables the proposed method to outperform state-of-the-arts in the KITTI benchmark.

Strengths: - The NAS for the volumetric stereo matching was presented, showing the top 1 accuracy in KITTI 2012 and 2015. - Searching for both cell-level and network-level structures leads to better performance.

Weaknesses: - The contribution is rather limited. The cell-level and network-level search are from [17] for semantic segmentation. The sole difference is to apply this two-level search strategy to the feature net and matching net, respectively. Though the candidate operations are reduced empirically and the residual cell are used within the cell level search, these are somehow incremental.

Correctness: - It would be better to clarify the proposed components. For instance, eq (3), bi-level optimization, and the use of two disjoint training sets are all from [17]. Though [17] was cited inside the text, adding this part inside 'Our Method’ seems inappropriate.

Clarity: The paper is well written and easy to understand.

Relation to Prior Work: The difference from previous works was mentioned, but it seems that the difference is not so significant.

Reproducibility: No

Additional Feedback: By extending [17] into volumetric stereo matching task, this work achieves the top-1 accuracy in KITTI 2012 and 2015 benchmark. Despite such outstanding performance, the technical contribution seems not so significant. The candidate operations are reduced experimentally and the residual cell is newly used, but these are somehow marginal. The joint search of the feature net and matching net leads to the performance gain according to Table 4, but this is not enough. Nevertheless, considering its outstanding performance, I will decide as ‘Marginally below the acceptance threshold'.


Review 3

Summary and Contributions: The paper introduces a first end-to-end cost volume-based NAS framework for stereo matching. The structure search is performed on two parts, feature net and matching net, with cell level search and network level search. The proposed LEAStereo ranked first on KITTI 2012 and 2015 benchmark and performs well on Middlebury 2014 datasets, which shows the effectiveness of automatic search compared to handcrafted design.

Strengths: 1, This paper proposes the first cost volume-based NAS framework for stereo matching and achieves better results even than handcrafted SOTA methods, which shows the advantages of NAS on dense pixel prediction task. 2, The authors provide necessary theoretical derivation for searching method of cell level search space.

Weaknesses: 1, I have concerns about the parameter β in network level search space. How it is trained? Is there a formula to describe it, like the parameter α in Eq. (2)? 2, I think it is better to compare with AANet [25] on the network structure and number of parameters. For structure, the construction of matching net is similar to AA module in AANet except for that matching net is searched automatically. For number of parameters, what is the advantages or disadvantages of LEAStereo?

Correctness: In line 64-65, the authors claim the several volumetric networks require 6-8 G of GPU memory for training per batch. I think the author has exaggerated this statement. In fact, for PSMNet [10], 4G of GPU memory is enough for training per batch.

Clarity: Yes, I think the paper is well written.

Relation to Prior Work: I think it has been mostly discussed except for the differences from AANet.

Reproducibility: Yes

Additional Feedback: Please refer to weakness and correctness above. -----Post Rebuttal Comments------ I have read the comments from other reviews and the rebuttal from authors. The rebuttal is reasonable and addresses my concerns. I agree with other reviews that this paper is not innovative enough since the techniques used in this paper are existing methods. But the impressive performance makes the paper acceptable. Therefore, I keep my original decision for this submission.


Review 4

Summary and Contributions: The authors propose to apply recent AutoML techniques to the Stereo depth estimation problem to find a new network architecture optimized for this task. In particular they use DART for the cell search and the hierarchical approach of AutoDeepLab for the overall network architecture. Human knowledge and previously developed architecture choices are used to limit the search space of possible architectures and make the problem tractable. The result of this effort is a new state-of-the-art architecture across all the major datasets for stereo depth estimation.

Strengths: + The results achieved by the method are without doubt impressive and represent the strongest achievement of the work. The results on Middlebury are particularly impressive for a deep learning based approach. + I found quite interesting the use of established architecture design choices, namely feature extraction followed by cost volume creation and refinement, to limit the search space and guide the autoML algorithm. This contribution seems to be crucial to obtain better performances than the previously published method based on AutoML. + The found architecture does not sacrifice speed for accuracy and is able to maintain a reasonable inference time (on GPU)

Weaknesses: - The paper is not particularly novel or exciting since it takes algorithms already applied in the field of semantic segmentation and applies them to the stereo depth estimation problem. The idea of using AutoML for stereo is not particularly novel either, as stated by the authors themselves, even if the proposed algorithm outperforms the previous proposal. - From my point of view the main reason to use AutoML approaches, besides improving raw performances, is extracting hints that can be reused in the design of new network architectures in the future. Unfortunately the authors did not spend much time commenting on these aspects. For example, what might be the biggest takeaways from the found architecture? - I would have liked to see an additional ablation study to better highlight the contribution of the proposed method with respect to AutoDispNet. The main differences with respect to the previously published work is the search performed also on the network level and the use of two separate feature and matching networks. Ablating the contribution of one and the other might have been interesting. - The evaluation on Middleburry should include for fairness a test of the found architecture running at quarter resolution to match the testing setup of all the other deep architecture. While it is true that the ability to run at higher resolution is an advantage of the proposed method there is nothing (besides hw limitation) preventing the other networks to run at higher resolution as well. Therefore I think that a fair comparison between networks running on the same test setup will improve the paper highlighting the contribution of the proposed method. - Some minor implementation details are missing from the paper, I will expand this point in my questions to the authors.

Correctness: I believe the claim to be correct and the experimental evaluation methodology to be correct.

Clarity: The paper is well written and easy to follow. Some minor implementation details can be made more clear, but this can be addressed in a polishing of the paper before camera ready.

Relation to Prior Work: Previous works have mostly been properly cited, however I found that some of them have not been properly presented or summarized in the related work section. For example all competing methods should be briefly introduced in the related work. I didn’t particularly like the placement of the related work section at the end of the paper, but this is a subjective judgement.

Reproducibility: Yes

Additional Feedback: I’m giving this paper a rating of 7 because I consider this work a very good improvement over current state of the art, even if the novelties introduced are relatively few. I have some questions that I would like to ask to the authors, and that i suggest to clarify in a revised version of the paper: (a.) The rescaling to ⅓ of the resolution is a little bit weird, most architectures proceed by powers of two, are there any specific reason to start downscaling the image by one third or is it just an experimental finding? (b.) For the Kitti 12 and 15 submissions have you used the same network? Or have you fine tuned two separate models? (c.) For the Middleburry submission, has the model been fine tuned? Or are the weights trained only on the SceneFlow dataset? ~~~ Post Rebuttal Comments ~~~ The authors have adressed most of my doubts on the rebuttal, therefore I keep my original acceptance rating for this submission.

[Author Response · NeurIPS 2020]

We thank all reviewers for their valuable comments. Below, we address their main concerns by quoting the comment followed by our response.

**R2: Q1. Use the same architecture for all datasets:** We indeed did this. To be precise, we only performed the architecture search once on the SceneFlow dataset and fine-tuned the weights on each benchmark separately. This implies the generalization capability of our proposal to a great degree. We conjecture that the main reason here is the use of a refined search space in our algorithm. In words, rather than growing the search space blindly in the hope of finding a good architecture, we have used the task-specific physics and inductive bias to constrain the search space.

**R2: Q2. Similar approach in different domains:** We agree that our design is tailored for the task of stereo matching and cannot be considered as a domain agnostic solution for a different problem. For different domains, better task-dependent NAS mechanisms are still a suitable solution to efficiently incorporate inductive bias and physics of the problem into the search space. We will reflect this in a revised version of our paper.

**R2: Q3. Interpretability:** We will tighten up our language based on your comment.

**R2: Q4. Better performance on large error thresholds:** We agree with the reviewer that the downsampling operations might prohibit the network to learn sub-pixel accuracy. It might also because of the loss function that does not encourage sub-pixel accuracy. We will acknowledge the issue (Bad 1.0 and 2.0 errors) and discuss accordingly in a revised version of our paper.

**R4: Q1. Technical Contribution:** Our method is the first NAS based method that can successfully do a *full architecture* search for an end-to-end stereo matching network. Note that directly applying [17] to stereo matching for a full architecture search is not viable (due to the huge memory requirements for high-resolution dense predictions, it can only search networks with limited layers). Also, as acknowledged by other reviewers, NAS has shown great success in classification tasks while not been very effective for dense prediction tasks yet. In our paper, we have successfully demonstrated that our NAS methods can achieve better performance than human-designed architectures by ranking 1 on various stereo matching benchmarks. We will revise the text to make this more clear. We will also release our code to ensure the reproducibility of the work and to improve this field.

**R5: Q1. Formula for updating $\beta$:** The parameter $\beta$ is updated similar in spirit to that of $\alpha$. Specifically, the following formula is used to update $\beta$, where $k$ represents the downsampling rate.

$$\boldsymbol{s}_l = \beta^l_{\frac{k}{2}\to k}\mathcal{C}(\boldsymbol{s}_{\frac{k}{2},l-1}, \boldsymbol{s}_{k,l-2};\alpha) + \beta^l_{k\to k}\mathcal{C}(\boldsymbol{s}_{k,l-1}, \boldsymbol{s}_{k,l-2};\alpha) + \beta^l_{2k\to k}\mathcal{C}(\boldsymbol{s}_{2k,l-1}, \boldsymbol{s}_{k,l-2};\alpha), \tag{1}$$

$$\beta^l_{\frac{k}{2}\to k} + \beta^l_{k\to k} + \beta^l_{2k\to k} = 1 \quad \text{and} \quad \beta^l \geq 0, \;\; \forall l, k. \tag{2}$$

**R5: Q2. Compare with AANet:** AANet(CVPR20) was officially published in June, 2020, after the deadline of NeurIPS. Structure-wise, the difference between our solution and AANet is that AANet builds multiple multi-scale cost volumes and processes them with 2D convolutions while our method constructs a feature volume and processes it with 3D convolutions. Our method benefits from fewer parameters (1.8M *vs* 3.9M) while enjoying higher performances (KITTI12 1.45% *vs* 2.04%, KITTI15 1.65% *vs* 2.03%, Middlebury 2.75% *vs* 10.8%). Per R5's comment, we will include AANet in a revised version of our paper.

**R6: Q1. Takeaways from the found architecture:** 1. The feature net does not need to be too deep to achieve good performance; 2. Larger feature volumes lead to better performance ($1/3$ is better than $1/6$); 3. A cost volume of $1/6$ resolution seems proper for good performance; 4. Multi-scale fusion seems important for computing matching costs (*i.e.* using a DAG to fuse multi-scale information). We will add a discussion about it in a revised version of our paper.

**R6: Q2. Our method *vs* AutoDispNet:** AutoDispNet has a very different network design philosophy than ours. It is a large U-Net-like architecture and tries to directly regress disparity maps from input images in pixel space (in contrast to our design which benefits from a feature and matching networks). Table on the right provides a head-to-head comparison. We will include this along a discussion in a revised version of our work.

| | Search Level | Params | KITTI 2012 | KITTI 2015 | Runtime |
|---|---|---|---|---|---|
| AutoDispNet | Cell | 111M | 1.70% | 2.18% | 0.9 s |
| LEAStereo | Full Network | 1.8M | 1.13% | 1.65% | 0.3 s |

Table 1: AutoDispNet *vs* LEAStereo

**R6: Q3. Test Middlebury on quarter resolution:** We would have liked to test Middlebury on a quarter resolution, but due to the strict submission policy, every method can only be submitted once. We are in the middle of negotiating the issue with the organizers of the Middlebury challenge to see if we can have another submission at the time of writing the rebuttal. Once we have it, we will add the results to the revised paper.

**R6: Q4. Minor implementation details:** A similar choice was also considered in other papers (*e.g.*, GA-Net). In comparison to $1/4$, the downsampling to $1/3$ will remove the need of upsampling twice. For question (b) and (c), please refer to the first question of R2.

[Meta-Review · NeurIPS 2020]

This paper initially received scores of 6,5,7, and 7. After the rebuttal R4 revised up from a 5 to a 6. The consensus from the reviewers was that while the technical novelty of the paper is not extremely high the results are important as neural architecture search for dense correspondence problems is under explored. Reviewers commented on the strong empirical performance for the same model across multiple datasets which is an important selling point for the paper. The authors are strongly encouraged to update the final paper to clarify the questions raised in the rebuttal - specifically the responses to R2's questions and the additional comparisons to AANet.